# The Effect of Pleural Effusion on Prognosis in Patients with Non-Small Cell Lung Cancer Undergoing Immunochemotherapy: A Retrospective Observational Study

**DOI:** 10.3390/cancers14246184

**Published:** 2022-12-14

**Authors:** Tomoka Nishimura, Eiki Ichihara, Toshihide Yokoyama, Koji Inoue, Tomoki Tamura, Ken Sato, Naohiro Oda, Hirohisa Kano, Daizo Kishino, Haruyuki Kawai, Masaaki Inoue, Nobuaki Ochi, Nobukazu Fujimoto, Hirohisa Ichikawa, Chihiro Ando, Katsuyuki Hotta, Yoshinobu Maeda, Katsuyuki Kiura

**Affiliations:** 1Department of Hematology and Oncology, Okayama University Graduate School of Medicine, Dentistry, and Pharmaceutical Sciences, Okayama 700-8558, Japan; 2Department of Allergy and Respiratory Medicine, Okayama University Hospital, Okayama 700-8558, Japan; 3Department of Respiratory Medicine, Ohara Healthcare Foundation, Kurashiki Central Hospital, Kurashiki 710-8602, Japan; 4Department of Respiratory Medicine, Ehime Prefectural Central Hospital, Matsuyma 790-0024, Japan; 5Department of Respiratory Medicine, NHO Iwakuni Clinical Center, Iwakuni 740-8510, Japan; 6Department of Respiratory Medicine, National Hospital Organization Okayama Medical Center, Okayama 701-1192, Japan; 7Department of Internal Medicine, Fukuyama City Hospital, Fukuyama 721-8511, Japan; 8Department of Respiratory Medicine, Japanese Red Cross Okayama Hospital, Okayama 700-8607, Japan; 9Department of Respiratory Medicine, Japanese Red Cross Society Himeji Hospital, Himeji 670-8540, Japan; 10Department of Internal Medicine, Okayama Saiseikai General Hospital, Okayama 700-0021, Japan; 11Department of Chest Surgery, Shimonoseki City Hospital, Shimonoseki 750-8520, Japan; 12Department of General Internal Medicine 4, Kawasaki Medical School, Okayama 700-8505, Japan; 13Department of Respiratory Medicine, Okayama Rosai Hospital, Okayama 702-8055, Japan; 14Department of Respiratory Medicine, KKR Takamatsu Hospital, Takamatsu 760-0018, Japan; 15Center for Innovative Clinical Medicine, Okayama University Hospital, Okayama 700-8558, Japan

**Keywords:** pleural effusion, non-small cell carcinoma, immune checkpoint inhibitors

## Abstract

**Simple Summary:**

Minimal data exists on pleural effusion (PE) for non-small cell lung cancer (NSCLC) patients undergoing combined ICI and chemotherapy. We retrospectively investigated how PE affects survival outcomes in patients with NSCLC undergoing this combined therapy. We identified 478 patients who underwent combined ICI therapy and chemotherapy; 357 patients did not have PE, and 121 patients did have PE. Patients with PE had significantly shorter progression-free survival and overall survival than those without PE. In addition, bevacizumab-containing regimens did not improve the survival outcomes for patients with PE. In conclusion, PE was associated with poor outcomes among patients with NSCLC undergoing combined ICI therapy and chemotherapy.

**Abstract:**

Objectives: Combined immune checkpoint inhibitor (ICI) therapy and chemotherapy has become the standard treatment for advanced non-small-cell lung cancer (NSCLC). Pleural effusion (PE) is associated with poor outcomes among patients with NSCLC undergoing chemotherapy. However, minimal data exists on PE for patients undergoing combined ICI and chemotherapy. Therefore, we investigated how PE affects survival outcomes in patients with NSCLC undergoing this combined therapy. Methods: We identified patients with advanced NSCLC undergoing chemotherapy and ICI therapy from the Okayama Lung Cancer Study Group–Immune Chemotherapy Database (OLCSG–ICD) between December 2018 and December 2020; the OLCSG–ICD includes the clinical data of patients with advanced NSCLC from 13 institutions. Then, we analyzed the treatment outcomes based on the presence of PE. Results: We identified 478 patients who underwent combined ICI therapy and chemotherapy; 357 patients did not have PE, and 121 patients did have PE. Patients with PE had significantly shorter progression-free survival (PFS) and overall survival (OS) than those without PE (median PFS: 6.2 months versus 9.1 months; *p* < 0.001; median OS: 16.4 months versus 27.7 months; *p* < 0.001). The negative effect of PE differed based on the patient’s programmed cell death-ligand 1 (PD-L1) expression status; with the effect being more evident in patients with high PD-L1 expression. In addition, PFS and OS did not differ between patients who did and did not undergo bevacizumab treatment; thus, bevacizumab-containing regimens did not improve the survival outcomes for patients with PE. Conclusion: PE is associated with poor outcomes among patients with NSCLC undergoing combined ICI therapy and chemotherapy.

## 1. Introduction

Immune checkpoint inhibitor (ICI) therapies have dramatically changed the treatment regimens for advanced non-small-cell lung cancer (NSCLC), and are now indispensable [1,2,3,4,5,6,7]. Programmed cell death-ligand 1 (PD-L1) expression is a predictive biomarker for a patient’s response to ICI therapy. However, the information it provides is incomplete, since certain patients with NSCLC and high PD-L1 expression do not respond to ICI. Additionally, several factors have been associated with ICI treatment outcomes, such as performance status (PS) [8], body mass index [9], previous use of antibiotics [10], and serological indicators (e.g., white blood cell count, neutrophil–lymphocyte ratio, and albumin level) [11].

Pleural effusion (PE) often accompanies advanced NSCLC, and has been associated with poor treatment outcomes [12,13]. However, these studies were conducted before the ICI therapy era. Some have reported significantly worse survival among patients with PE treated with ICI monotherapy compared to those without PE [14,15], suggesting that PE negatively affects the prognosis of this patient population. However, overall, data on the associations between PE and ICI therapy outcomes for patients with NSCLC are lacking, and minimal data exists on PE for patients undergoing combined ICI and chemotherapy [16]. Thus, the effects of PE on patients with NSCLC undergoing a combined treatment regime remain unclear. Therefore, this study used the Okayama Lung Cancer Study Group–Immune Chemotherapy Database (OLCSG–ICD), which includes the clinical data of patients with advanced NSCLC, to retrospectively analyze the effect of PE on treatment outcomes.

## 2. Methods

### 2.1. Patients

We performed a branch study using data from the OLCSG–ICD database from December 2018 to December 2020; the database contains the clinical data of consecutive patients with NSCLC who started first-line systemic therapy (except for molecular targeted therapy) to treat advanced NSCLC from 13 institutions. Each institution’s review board approved this study.

We compared the treatment outcomes of patients with NSCLC and malignant PE to those without PE. PE malignancy was not always confirmed by pathological examination and was clinically diagnosed in some cases. The overall survival (OS), progression-free survival (PFS), disease control rate (DCR), and objective response rate (ORR) were evaluated following the Response Evaluation Criteria in Solid Tumors criteria version 1.1. PFS was defined as the time from diagnosis of incurable advanced lung cancer to disease progression or death from any cause. OS was defined as the time from diagnosis of incurable advanced lung cancer to death from any cause.

### 2.2. Statistical Analyses

Differences in patient characteristic between the groups were analyzed using Fisher’s exact test. PFS was defined as the time from ICI therapy initiation to disease progression or death, and OS was defined as the time from ICI therapy initiation to death. These analyses were performed using the Kaplan-Meier method and log-rank test. Multivariate analysis was performed using the Cox proportional hazards model with the variables. DCR was defined as the sum of the complete response (CR), partial response (PR), and stable disease, and ORR was defined as the sum of the CR and PR. Statistical analyses were conducted using STATA software program version 11.0 (Stata, College Station, TX, USA), and *p*-values of <0.05 were considered statistically significant.

## 3. Results

### 3.1. Patients Characteristics

This study included 478 patients; 121 patients had PE, and 357 did not (Table 1). The patient characteristics did not differ between those with and without PE, except for the ICI agent (pembrolizumab or atezolizumab). The median age was 69/70 years; 84.3% and 80.1% of patients with and without PE were men, respectively. Most patients had a PS of 0 to 1 (PE: 91.6%; no PE: 86.8%) and smoking history (PE: 84.1%; no PE: 86.8%). Moreover, 79.3% and 75.6% with and without PE had non-squamous carcinoma, respectively.

The PD-L1 expression status was obtained for 95 patients with PE (78.5%) and 311 patients without PE (87.1%). Of these, 22 (18.2%) and 85 (23.8%) patients with and without PE had high PD-L1 expression (PD-L1 50% or more), respectively. Furthermore, 9 (7.4%) and 29 (8.1%) patients with and without PE had epidermal growth factor receptor (i.e., EGFR) or anaplastic lymphoma kinase (i.e., ALK) gene alterations. Finally, patients with PE used atezolizumab more frequently than those without PE (PE: 21.5%; no PE: 13.2%, *p* = 0.04), likely reflecting a more frequent use of the bevacizumab-containing regimen (i.e., atezolizumab, bevacizumab, carboplatin, and paclitaxel). Consistently, patients with PE used bevacizumab more frequently than those without PE (PE: 18.2%; no PE: 9.0%, *p* = 0.008).

### 3.2. PE Negatively Affects the Combined ICI Therapy and Chemotherapy Treatment Outcomes

Table 2 presents the response outcomes. Patients with PE had significantly worse DCR and ORR than those without PE (DCR: 86.0% vs. 93.9%, *p* = 0.016; ORR: 44.7 vs. 62.8%, *p* = 0.001). The median PFS was 6.2 months for patients with PE (95% confidence interval [CI]: 4.7–8.6) and 9.1 months for those without PE (95% CI: 8.0–11.0; hazards ratio [HR]: 1.40, 95% CI: 1.08–1.81, *p* < 0.001; Figure 1A). The median OS durations for patients with and without PE were 16.4 months (95% CI: 12.5–22.1) and 27.7 months (95% CI: 23–not reached), respectively (HR: 2.12, 95% CI: 1.56–2.89, *p* < 0.001; Figure 1B). Multivariate analysis identified that PE, a poor PS, and low PD-L1 expression (<50%) were independently associated with a shorter PFS and OS (Table 3).

### 3.3. The Negative Effect of PE Differed Based on PD-L1 Expression

We used the PD-L1 expression status to investigate the relationship between PE and PD-L1 expression (Figure 2). Among patients with high PD-L1 expression, PFS and OS were significantly shorter for those with PE than for those without PE (PFS: HR: 3.24, 95% CI: 1.73–6.03, *p* < 0.001; OS: HR: 3.06, 95% CI: 1.50–6.24, *p* = 0.002; Figure 2A,B). Among patients with low PD-L1 expression, PFS did not differ between those with and without PE (HR 1.34, 95% CI: 0.99–1.83, *p* = 0.080); OS differed significantly, but it was less apparent (OS: HR: 1.95, 95% CI: 1.34–2.84, *p* < 0.001; Figure 2C,D). These data suggest that PE has a greater negative effect when patients with NSCLC have high PD-L1 expression.

### 3.4. The Bevacizumab-Containing Regimen Did Not Improve Survival Outcomes in Patients with PE

Among those with PE, we investigated the outcomes based on their treatment regimen: bevacizumab-containing regimen (i.e., a combination of bevacizumab, carboplatin, paclitaxel, and atezolizumab) or other (i.e., a non-bevacizumab regimen). Patient characteristics of the patients are listed in Appendix A. Among the 121 patients with PE, 22 patients were treated with the bevacizumab-containing regimen, and the remaining 99 were treated with other regimens. There were significantly more patients with *EGFR* or *ALK* gene alteration in those treated with bevacizumab, likely reflecting potential efficacy of the bevacizumab-containing regimen (i.e., atezolizumab, bevacizumab, carboplatin, and paclitaxel) for the treatment of NSCLC with *EGFR* or *ALK* gene alteration [6,17]. The patient characteristics did not differ between those with and without bevacizumab, except for the *EGFR*/*ALK* gene mutation status and the ICI agent (pembrolizumab or atezolizumab). PFS and OS did not differ between patients receiving the bevacizumab-containing regimen and those receiving other regimens (Figure 3).

### 3.5. Pleural Intervention Did Not Improve Survival Outcomes in Patients with PE

Finally, we examined whether pleural interventions, such as drainage and pleurodesis prior to initiating systemic therapy, would affect patient outcomes. Among the 121 patients with PE, 66 patients received drainage and/or pleurodesis and 51 did not. The remaining four patients lacked the data regarding pleural interventions. PFS and OS did not differ between patients with pleural intervention and those without it (Appendix A). The median PFS was 6.7 months for patients with pleural intervention (95% CI: 4.4–8.7) and 6.7 months for those without the intervention (95% CI: 3.5–9.8, *p* = 0.730; Appendix AA). The median OS durations for patients with and without pleural intervention were 17.5 months (95% CI: 9.9–not reached) and 13.2 months (95% CI: 11.6–22.1), respectively (*p* ≤ 0.189; Appendix AB).

## 4. Discussion

This study investigated the effects of PE on the treatment outcomes of patients with NSCLC undergoing combined ICI therapy and chemotherapy. Some studies have reported that PE negatively affects the prognosis of patients with NSCLC undergoing ICI monotherapy [14,15]. However, none have investigated these effects in patients undergoing combined therapy.

Patients with malignant PE have high serum vascular endothelial growth factor (VEGF) levels [18,19]. VEGF promotes PE by increasing vascular permeability; it also promotes fluid accumulation and activates myeloid-derived suppressor cells, resulting in immunosuppression of the tumor microenvironment [20,21]. In this study, patients with PE had worse survival outcomes, potentially attributed to immunosuppression from increased VEGF levels.

In this study, patients with PE received atezolizumab significantly more frequently than patients without PE (Table 1), indicating that they received the regimen comprising atezolizumab, bevacizumab, carboplatin, and paclitaxel. Bevacizumab is a humanized monoclonal antibody that inhibits VEGF from binding to its receptor on vascular endothelial cells. VEGF signaling pathway inhibition normalizes the immature structure of tumor vessels, decreases vascular permeability, and reduces PE [22,23,24]. Thus, a bevacizumab/chemotherapy combination has been highly effective for patients with NSCLC and PE [22,23]. However, in this study, bevacizumab combined with ICI therapy and chemotherapy did not improve the survival outcomes. Therefore, bevacizumab’s efficiency against PE might be counteracted when combined with ICI agents, since ICIs may cause PE as an immune-related adverse event [25].

PE is a known poor prognostic marker in patients with advanced NSCLC [26]. We found that PE negatively affected PFS only if the patient had high PD-L1 expression. This result suggests that PE may be a predictive marker for patients undergoing combined ICI therapy and chemotherapy.

This study has some limitations. Firstly, this was a retrospective study with heterogeneous data. Secondly, our database did not include the data of other metastatic sites, such as liver and pulmonary metastasis, which also negatively affect patients with NSCLC undergoing ICI therapy [27]. Thus, confounding bias is possible. Thirdly, PE was not diagnosed by pleural fluid cytology. Therefore, physician selection bias is possible. Finally, no suggestive data was obtained to improve outcomes of those with PE. Although we found PE was associated with poor outcomes among patients with NSCLC undergoing combined ICI therapy and chemotherapy, we could not find a solution to that. Bevacizumab or pleural intervention was noted as a potential factor that improved outcomes, but did not show a survival advantage. Further research is needed to resolve this issue.

## 5. Conclusions

Our study indicates that PE is a poor prognostic factor for patients with NSCLC undergoing combined ICI therapy and chemotherapy treatment. Furthermore, concomitant anti-VEGF antibody therapy with bevacizumab did not improve the outcomes of patients with NSCLC and PE. Therefore, treatment strategies should be established based on the patient’s PE status.

## Figures and Tables

**Figure 1 cancers-14-06184-f001:**
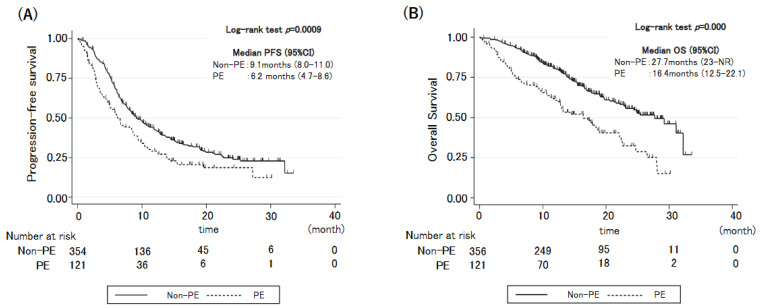
Kaplan-Meier curves for (**A**) progression-free survival and (**B**) overall survival based on the patient’s pleural effusion status.

**Figure 2 cancers-14-06184-f002:**
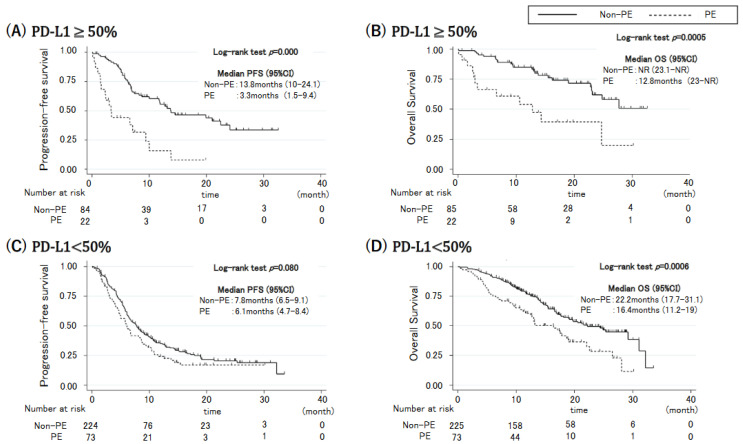
Kaplan-Meier curves for progression-free survival and overall survival in patients with (**A**,**B**) high (≥50%) and (**C**,**D**) low (<50%) programmed cell death-ligand 1 (PD-L1) expression.

**Figure 3 cancers-14-06184-f003:**
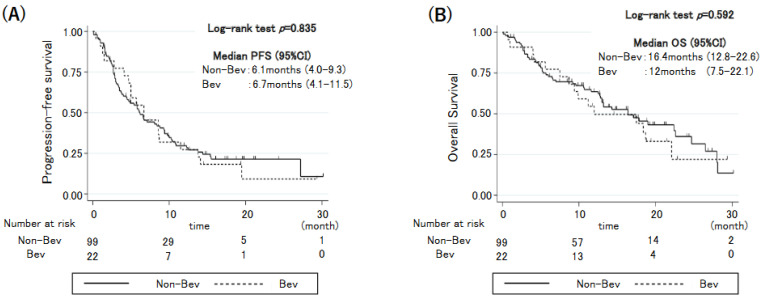
Kaplan-Meier curves for (**A**) progression-free survival and (**B**) overall survival in patients undergoing immunochemotherapy with (Bev) and without (Non-Bev) bevacizumab.

**Table 1 cancers-14-06184-t001:** Patient characteristics.

	Non-PE	PE	
	*n* = 357	*n* = 121	*p*-value
Age (years)			
median (range)	69 (34–82)	70 (42–84)	
≥70	171 (47.9%)	67 (55.4%)	0.172
<70	186 (52.1%)	54 (44.6%)	
Sex			
female	71 (19.9%)	19 (15.7%)	0.348
male	286 (80.1%)	102 (84.3%)	
Performance Status			
0–1	327 (91.6%)	105 (86.8%)	0.152
≥2	30 (8.4%)	16 (13.2%)	
Smoking History			
never	56 (15.9%)	16 (13.2%)	0.558
current or former	297 (84.1%)	105 (86.8%)	
Histology			
squamous	87 (24.4%)	25 (20.7%)	0.457
non-squamous	270 (75.6%)	96 (79.3%)	
PD-L1 expression			
<1%	116 (32.5%)	38 (31.4%)	0.130
1–49%	110 (30.8%)	35 (28.9%)	
≥50%	85 (23.8%)	22 (18.2%)	
Unknown	46 (12.9%)	26 (21.5%)	
*EGFR* or *ALK* gene alteration	29 (8.1%)	9 (7.4%)	0.554
Disease Stage			
advanced	278 (78.1%)	98 (81.0%)	0.606
rec	78 (21.9%)	23 (19.0%)	
ICI drug			
Pembrolizumab	310 (86.8%)	95 (78.5%)	0.04
Atezolizumab	47 (13.2%)	26 (21.5%)	
Bevacizumab-containing Regimen			
yes	32 (9.0%)	22 (18.2%)	0.008
no	325 (91.0%)	99 (81.8%)	

PE: pleural effusion, PD-L1: programmed death-ligand 1, EGFR: epidermal growth factor receptor, ALK: anaplastic lymphoma kinase, rec: postoperative or postchemoradiotherapy recurrence, ICI: immune checkpoint inhibitor.

**Table 2 cancers-14-06184-t002:** Response outcomes in Non-PE and PE groups.

	Non-PE	PE	
	*n* = 357	*n* = 121	*p*-value
Best response rate			
CR	9 (2.6%)	1 (0.9%)	
PR	205 (60.0%)	50 (43.9%)	
SD	107 (31.3%)	47 (41.2%)	
PD	21 (6.1%)	21 (6.1%)	
Unknown	15	7	
Disease control rate			
yes	321 (93.9%)	98 (86.0%)	0.016
no	21 (6.1%)	16 (14.0%)	
Objective response rate			
yes	214 (62.8%)	51 (44.7%)	0.001
no	128 (37.4%)	63 (55.3%)	

PE: pleural effusion, CR: complete response, PR: partial response, SD: stable disease, PD: progressive disease.

**Table 3 cancers-14-06184-t003:** Multivariate analysis of factors associated with progression-free survival and overall survival.

	PFS		OS	
	HR (95% CI)	*p*	HR (95% CI)	*p*
Pleural effusion				
No	1.0		1.0	
Yes	1.50 (1.14–1.97)	0.004	2.09 (1.50–2.92)	<0.001
Age (years)				
≥70	1.0		1.0	
<70	1.08 (0.85–1.39)	0.497	1.08 (0.79–1.48)	0.604
Sex				
female	1.0		1.0	
male	1.24 (0.83–1.85)	0.277	1.21 (0.72–2.03)	0.1466
Performance status				
0–1	1.0		1.0	
≥2	2.32 (1.60–3.38)	<0.001	2.83 (1.87–4.30)	<0.001
Smoking History				
never	1.0		1.0	
current or former	0.91 (0.59–1.42)	0.707	1.33 (0.71–2.50)	0.363
Histology				
squamous	1.0		1.0	
non-squamous	0.64 (0.48–0.85)	0.002	0.76 (0.53–1.09)	0.150
PD-L1 expression				
<50%	1.0		1.0	
≥50%	0.64 (0.48–0.86)	0.004	0.65 (0.45–0.95)	0.029
ICI drug				
Pembrolizumab	1.0		1.0	
Atezolizumab	0.76 (0.36–1.64)	0.499	0.24 (0.34–1.77)	0.165
Bevacizumab-containing regimen				
Yes	1.0		1.0	
No	1.86 (0.81–4.28)	0.143	4.49 (0.59–34.00)	0.146

PFS: progression-free survival, OS: overall survival, HR: hazard ratio, CI: confidence interval.

## Data Availability

The datasets generated and/or analyzed during the current study are available from the corresponding author on reasonable request.

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
