# Peer review of "The Effect of Pleural Effusion on Prognosis in Patients with Non-Small Cell Lung Cancer Undergoing Immunochemotherapy: A Retrospective Observational Study"

_cancers, 2022, doi:10.3390/cancers14246184_

Round 1

Reviewer 1 Report

The authors retrospectively assessed whether pleural effusion negatively affects the efficacy of chemoimmunotherapy.  They found patients with PE had significantly shorter PFS and OS compared to patients without PE. They also reported that the addition of bevacizumab to chemoimmunotherapy did not improve the outcome of patients with PE. Although the mechanism by which adversely affects prognosis is unknown, these findings provide useful information for clinical physicians treating lung cancer patients.

Minor comments

1.       Did the authors collet data from consecutive patients?

2.       In table 1, the titles of group; “Non-PE” “PE”, are misaligned.

3.       The first paragraph on page 5, the authors reported PFS a OS with high PD-L1 expression and PFS and OS with low PD-L1 expression in reverse.

4.       w/wo Bev should be added on table 1 and table 3 if the authors emphasize on the effect of BEV.

5.       In Table 3, smoking status, metastatic site such as brain metastasis should be added.

6.       Table of patient characteristics by w/wo BEV are needed even in suppl figure/table if the authors show figure 3. 

7.       The control rate of pleural effusion could be useful information.

8.       The impact of Pleural intervention, such as w/wo drainage or pleurodesis, on prognosis should be evaluated.

9.       Authors should cite the report below.

JTO Clin Res Rep. 2022 Jun 3;3(7):100355.

Author Response

Response to Reviewer 1

The authors retrospectively assessed whether pleural effusion negatively affects the efficacy of chemoimmunotherapy.  They found patients with PE had significantly shorter PFS and OS compared to patients without PE. They also reported that the addition of bevacizumab to chemoimmunotherapy did not improve the outcome of patients with PE. Although the mechanism by which adversely affects prognosis is unknown, these findings provide useful information for clinical physicians treating lung cancer patients.

We really appreciate your thoughtful and important comments. We change the points that you commented.

Minor comments

  1. Did the authors collet data from consecutive patients?

REPLY Yes, we collected the data from the consecutive patients. We clarify this point in Methods (page 2, first paragraph of “2.1. Patients”).

  1. In table 1, the titles of group; “Non-PE” “PE”, are misaligned.

REPLY We thank the reviewer for pointing that out. We revise the alignment of the table.

  1. The first paragraph on page 5, the authors reported PFS a OS with high PD-L1 expression and PFS and OS with low PD-L1 expression in reverse.

REPLY We thank the reviewer again for pointing that out. We revise the corresponding part.

  1. w/wo Bev should be added on table 1 and table 3 if the authors emphasize on the effect of BEV.

REPLY According to the comment, we add w/wo Bev on table 1 and table 3, and describe that in Results (page 3, second paragraph of “3.1. Patients Characteristics”).

  1. In Table 3, smoking status, metastatic site such as brain metastasis should be added.

REPLY We think this is an important point. Unfortunately, we do not have the detailed data of each metastatic cite in the database but have the data of smoking status. Therefore, we conducted the multivariate analysis again including smoking status and found the consistent data that PE is an independent poor survival factor (Table 3).

  1. Table of patient characteristics by w/wo BEV are needed even in suppl figure/table if the authors show figure 3. 

REPLY We totally agree with the comment. We add patient characteristics by w/wo BEV as supplementary Table 1 and describe that in Results (page 6, first paragraph of “3.4. The Bevacizumab-containing Regimen Did Not Improve Survival Outcomes in Patients With PE”).

  1. The control rate of pleural effusion could be useful information.

     REPLY We agree that the control rate of PE is useful. However, we unfortunately do not have the data regarding PE control rate in our database. 

  1. The impact of Pleural intervention, such as w/wo drainage or pleurodesis, on prognosis should be evaluated.

     REPLY We appreciate the important comment. As the reviewer suggested, we investigated the survivals of the patients according to existence of pleural interventions. There was no significant difference of survivals between intervention group and nonintervention group. We add the results as supplementary Figure 1 and describe that as “3.5. Pleural Intervention Did Not Improve Survival Outcomes in Patients With PE” in the second paragraph of page 6.

  1. Authors should cite the report below.

JTO Clin Res Rep. 2022 Jun 3;3(7):100355.

REPLY We thank the reviewer for letting us notice the paper. We cite it in our manuscript.

Reviewer 2 Report

Dear Authors,

Your manuscript titled "The effect of pleural effusion on prognosis in patients with non-small cell lung cancer undergoing immunochemotherapy: a retrospective observational study", is an interesting work with the research outcome indicating presence of pleural effusion (PE) is associated with poor outcomes among patients with NSCLC undergoing chemotherapy. 

Even though the authors have supported their findings with statistical data from the NSCLC database for 2-year period. there are elements that are left unanswered like the possibility of chemotherapy impact on the occurrence of PE and what advantage this information to the clinical community in improving the treatment options that will improve the overall survival of the patients.

if more impetrations on the impact of the other disease condition to occurrence of PE can help readers.

Author Response

Dear Authors,

Your manuscript titled "The effect of pleural effusion on prognosis in patients with non-small cell lung cancer undergoing immunochemotherapy: a retrospective observational study", is an interesting work with the research outcome indicating presence of pleural effusion (PE) is associated with poor outcomes among patients with NSCLC undergoing chemotherapy. 

Even though the authors have supported their findings with statistical data from the NSCLC database for 2-year period. there are elements that are left unanswered like the possibility of chemotherapy impact on the occurrence of PE and what advantage this information to the clinical community in improving the treatment options that will improve the overall survival of the patients.

if more impetrations on the impact of the other disease condition to occurrence of PE can help readers.

REPLY  We thank the reviewer from the bottom of my heart for the important and thoughtful comment. As the reviewer commented, our study did not provide any suggestion what strategy should be adopted for NSCLC patients with PE. To find other condition to improve outcomes of NSCLC with PE, we investigated whether interventions such as drainage or pleurodesis would improve the survivals. Unfortunately, those interventions did not improve the survivals.

We show the result as Supplementary Figure 1 and describe that as “3.5. Pleural Intervention Did Not Improve Survival Outcomes in Patients With PE” in Results (page 6). We also discuss this as a limitation of our study in Discussion (page 7).